# Stress and resilience during pregnancy: A comparative study between pregnant and non-pregnant women in Ethiopia

**Mubarek Abera**[1]*, **Charlotte Hanlon**[2,3,4], **Hikma Fedlu**[5], **Mary Fewtrell**[6], **Markos Tesfaye**[7], **Jonathan C. K. Wells**[6]

**1** Department of Psychiatry, Faculty of Medical Sciences, Jimma University, Jimma, Ethiopia, **2** Department of Health Services and Population Research, Centre for Global Mental Health, Institute of Psychiatry, Psychology and Neuroscience, King's College London, London, United Kingdom, **3** Department of Psychiatry, WHO Collaborating Centre for Mental Health Research and Capacity-Building, School of Medicine, College of Health Sciences, Addis Ababa University, Addis Ababa, Ethiopia, **4** Centre for Innovative Drug Development and Therapeutic Trials for Africa (CDT-Africa), College of Health Sciences, Addis Ababa University, Addis Ababa, Ethiopia, **5** Department of Public health officer, Faculty of Medical Sciences, Jimma University, Jimma, Ethiopia, **6** Population, Policy and Practice Research and Teaching Department, UCL Great Ormond Street Institute of Child Health, London, United Kingdom, **7** Department of Psychiatry, St. Paul's Hospital Millennium Medical College, Addis Ababa, Ethiopia

* mubarek.abera@ju.edu.et, abmubarek@gmail.com

**Data Availability Statement:** All required data are included in the manuscript.

**Funding:** CH receives support from the National Institute of Health Research through the NIHR

## Abstract

### Background

Stress during pregnancy is associated with perturbances in maternal psychology and physiology, and results in adverse pregnancy and birth outcomes. However, little attention has been given to understand maternal stress and its potential negative consequences in many low- and middle-income countries. We aimed to investigate whether pregnancy is associated with greater stress and lower psychological resilience among women living in Jimma, Southwest Ethiopia.

### Method

An institution-based comparative cross-sectional study design was implemented in Jimma University Medical Center and Jimma health centers from 15 September to 30 November 2021. Women attending antenatal care and family planning services were invited to participate in the study. Participants were interviewed using the Perceived Stress Scale (PSS-10), Brief Resilience Scale (BRS), distress questionnaire-5, and the Household Food Insecurity Access Scale (HFIAS). Linear regression analysis was used to test associations between pregnancy (exposure) and outcomes of interest (stress and resilience scores), while adjusting for potential confounders. Stress and resilience were mutually adjusted for one another in the final model.

### Results

A total of 166 pregnant and 154 non-pregnant women participated, with mean age of 27.0 SD 5.0 and 29.5 SD 5.3 years respectively. Pregnancy was associated with increased stress score by 4.1 points (β = 4.1; 95% CI: 3.0, 5.2), and with reduced resilience by 3.3

Global Health Research Group on Homelessness and Mental Health in Africa (NIHR134325) and the SPARK project (NIHR200842) using UK aid from the UK Government. The views expressed in this publication are those of the authors and not necessarily those of the NIHR or the Department of Health and Social Care. CH receives support from the Wellcome Trust through grants 222154/Z20/Z and 223615/Z/21/Z.

**Competing interests:** The authors have declared that no competing interests exist.

points ($\beta$ = -3.3; 95% CI: -4.5, -2.2) in a fully adjusted model. In mutually-adjusted models, pregnancy was independently associated with greater stress ($\beta$ = 2.9, 95% CI 1.8, 3.9) and lower resilience ($\beta$ = -1.3, 95% CI: -2.5, -0.2) compared to non-pregnant women.

## Conclusion

In this low income setting, pregnancy is associated with greater vulnerability in the mental health of women, characterized by greater perceived stress and diminished resilience. Context-relevant interventions to improve resilience and reduce stress could help improve the health and wellbeing of mothers, with potential benefits for their offspring.

## Introduction

The stress response, an adaptive component of physiology, represents a survival strategy during exposure to threats, adverse experiences or stressors in life [1]. When activated, the stress response prepares the body for 'fight or flight' reaction to promote safety and protection [1]. However, chronic activation of the stress response affects the body negatively and impairs health, wellbeing and performance [2]. The extent to which exposure to threats drives the perception of stress varies, however, it depends on an individual's ability to cope or adapt with stressors and successfully bounce back to the normal homeostasis from the effect of adversities [3]. The impact of stressors on the stress response system is mediated by coping and adaptive strategies, forms of resilience, that buffer the adverse effects on health and wellbeing [4]. The development of this psychological resilience is a dynamic process across the life span formed as a product of the interaction between biological, psychological and socio-environmental factors [5, 6]. People with different levels of resilience therefore respond differently to a similar set of stressors, such that those with low resilience are more prone to the adverse consequences of stress.

Women in low-income countries experience disproportionate levels of stressors related to household responsibilities, as well as gender inequalities such as an elevated risk of malnutrition, dietary inadequacy and violence (in particular intimate partner violence) [7]. Moreover, emerging evidence indicates that the state of pregnancy itself induces additional stress to pregnant women [8]. The transition in social role associated with becoming a mother [9] may be accompanied by new physiological sources of stress, and potentially greater sensitivity to stress. This issue has major implications for public health, as experiencing stress during pregnancy has consequences not only for the mother, but also for the offspring who may be exposed to the physiological signals of stress passing through the placenta [10–12].

Stress in pregnancy can affect maternal health/wellbeing and quality of life by triggering maladaptive emotional and physiological states. Because of its negative impact on maternal and fetal health and nutrition during pregnancy, maternal stress may lead to adverse pregnancy and birth outcomes, such as shorter gestational age, prolonged labor, abortion, stillbirth, low birth weight, congenital anomalies, maternal perinatal infections, preeclampsia and hemorrhage [2, 13, 14]. These associations tend to be of dose-response nature, whereby the greater the maternal stress, the greater the likelihood or magnitude of adverse outcomes. Perinatal complications and prolonged labor were also associated with antenatal common mental disorders in Ethiopian settings [15, 16].

In turn, children born to stressed mothers have increased risk of morbidity, growth restriction, and cognitive disability, and may have an elevated risk of mental and behavioral problems during childhood such as anxiety, depression and attention deficit disorders [2, 14, 17–21].

This scenario contributes to the persistent burden of stunted growth and development among children in LMICs [22]. Finally, adults who survived stress in prenatal and postnatal life may have higher risk of non-communicable diseases (NCDs) and reduced human capital [11]. In turn, this has implications for society, through reduced economic productivity [21, 23, 24]. In this way, stress during pregnancy may contribute to an inter-generational cycle of disadvantages [25, 26].

Research on prenatal stress is important especially in countries where poverty and adverse life circumstances are abundant. A recent WHO report indicated that maternal mental health is the missed component of maternal health in LMICs [27, 28]. To date, for example, there are no published data on stress and resilience during pregnancy in Ethiopia, a low-income country with a high burden of maternal mental health problems, child stunting and other adverse environments undermining optimal child development [29, 30]. The present study therefore aimed to investigate whether pregnancy is associated with higher levels of stress and lower resilience, by comparing these outcomes between pregnant and non-pregnant women in the city of Jimma, Ethiopia.

## Method and participants

### Setting

The study was conducted in the urban setting of Jimma Zonal City, Oromia Region, Ethiopia. The city has two governmental hospitals, two private hospitals, five health centers, and more than 10 private clinics. In addition, there are urban health extension workers providing maternal and child health care services in the city. The total population of the city based on the 2021 projection is estimated to 240,000. The study was conducted during the period between September and November, 2021.

### Design

We employed an institution-based comparative cross-sectional study design.

### Population

We recruited pregnant and non-pregnant women aged between 18 and 40 years, attending antenatal care and family planning services respectively at Jimma University Medical Center and Jimma health centers. We invited consecutive women attending antenatal care and family planning services to participate until we achieved the target sample size.

### Assessment

**Background characteristics.** Information on age, parity, marital status, social support, religion, family size, income level, educational and occupational status was collected using a structured self-report questionnaire.

**Stress.** Stress was assessed using the 10 item Perceived Stress Scale (PSS-10) questionnaire [31] which has been validated in the local context in Ethiopia [32]. PSS items were scored as 0 (Never), 1 (Almost Never), 2 (Sometimes), 3 (Fairly Often), and 4 (Very Often). Items 4, 5, 7 and 8 were positively phrased questions that required reverse coding during analysis. The sum score ranges from 0 (low level) to 40 (maximum level). (See Appendix A in S1 File)

**Resilience.** Psychological resilience was assessed using the Brief Resilience Scale (BRS-6) [33] which contained 6 items scored as 1 (strongly disagree), 2 (disagree), 3 (neutral), 4 (agree), and 5 (strongly agree) [33]. Item 2, 4 and 6 were negatively phrased and required

reverse coding during analysis. The sum score for resilience ranges from 6 (low resilience) to 30 (high resilience) points (See Appendix B in S1 File).

**Exposure.** For pregnancy status, women were classified as pregnant or non-pregnant as confirmed by an obstetricians or midwife nurses.

**Covariates.** Additional variables on the following parameters were collected.

**Household food insecurity access scale (HFIAS).** HFIAS validated for Ethiopian context was used [34].

**Social support.** Perceived social support was assessed using a single question enquiring if women currently have adequate or inadequate perceived social support.

**Physical activity.** Level of current physical activity was assessed using a single question enquiring if women's current level of physical activity is decreased, similar or increased as compared to their previous experience.

**Substance use.** History of life time use, last 12 months use, and current (last 3 months) substance (Khat, alcohol, Nicotine, Shisha, Marijuana, Cannabis and other) use were asked.

**Psychological distress.** Women were screened for psychological distress using the distress questionnaire which contain 5 items (DQ-5) [35].

**Method of data collection.** An interviewer-administered structured questionnaire was used for data collection. Two bachelor-level health professionals conducted the interview. To maintain quality of the data, the principal investigator closely supervised data collection. Data were checked for completeness on the day of collection.

## Data management and analysis

All data were entered into Epidata version 3.1 and transported to STATA-17 for analysis. The data were checked for appropriateness and completeness before entry and then visualized and cleaned using statistical software. Percentage and frequencies were used to describe categorical data while mean and standard deviation (SD) or median and inter quartile range (IQR) were used to describe continuous data depending on their distributions. The total scores of stress and resilience were computed by summing the response for individual items. For stress, a score of 0–13 was considered as low, 14–26 as moderate and 27–40 as high stress [36]. Individual resilience scores for the six items were summed to a total score ranging from 5–30, and then divided by 6 to obtain categorical level whereby values from 1–2.99 were considered low, 3.00 to 4.30 as normal, and 4.31–5.00 as high resilience [33]. T-test and Chi-square test were conducted to examine relation of stress and resilience with status of pregnancy. Bivariate and multivariate linear regression analysis were used to investigate the associations of stress and resilience with pregnancy, adjusting for confounders.

After checking assumptions for linear regression, five different regression models were developed and the outcome variables (stress and resilience) were regressed on the main exposure (pregnancy status) adjusted for different covariates. **Model 1:** unadjusted (outcome variables regressed on pregnancy); **Model 2:** Model 1 further adjusted for age; **Model 3:** Model 2 further adjusted for parity; **Model 4:** (Fully adjusted model): Model 3 further adjusted for marital status, social support, religion, family size, income, educational and occupational status, household food insecurity level, physical activity, substance use status, ever use of substance and psychological distress; **Model 5:** (stress and resilience mutually adjusted to each other)— *model 5a*: model 4 is further adjusted for resilience score and stress is the outcome variable; *model 5b*: model 4 is further adjusted for stress score and resilience is the outcome variable. The same regression models were repeated for each specific PSS-10 and BRS-6 items to identify items contributing for the overall association between pregnancy and the total stress score or resilience score. Life time substance use, last 12 month substance use and last 3 months

history of substance use were interchangeably entered in to the full regression model, but only life time history of substance use influenced the estimate for the main exposure compared to the other levels of substance use.

P-values and 95% confidence intervals (CI) were reported as measures of statistical significance and the magnitude of effect respectively. The data were presented using table and forest plot graphs.

## Ethics

This study was approved by the IRB of Jimma University. Participant safety, privacy, and confidentiality were ensured. All women were given information about the study and provided their written consent for voluntary participation. The right not to participate or to withdraw from the study was respected. All women during the study period were given equal opportunity to participate in the study. Women with high stress scores were linked to a counseling service.

## Results

### Background characteristics

A total of 327 women were invited and 320 (166 pregnant and 154 non-pregnant) women participated in the study. In pregnant compared to non-pregnant women, mean ages were 24.7 SD 5.1 years and 29.4 SD 5.3 years, average fertility rate was 2.0 SD 1.2 and 2.6 SD 1.1, and average family size was 3.6 SD 1.4 and 5.2 SD 1.3, respectively. Most of the participants were Muslim by religion, and married. Among the pregnant women, 15% illiterate and 41% had attended some level of primarily education, while the corresponding figures were only 1% and 66% in the non-pregnant women. In terms of social support, only 28% of pregnant and 49% of non-pregnant women reported having adequate social support. Most of the participants reported having a low level of physical activity, but the frequency reporting decreased physical activity was higher (63%) among the pregnant women than 32% in the non-pregnant women. All of these variables differed significantly between pregnant and non-pregnant women. See Table 1.

### Stress and resilience

Internal consistency for PSS-10 and BRS-6 scales were 0.7 and 0.7, respectively. The mean PSS score was 18.7 SD 4.3 and 14.4 SD 4.3 in pregnant and non-pregnant women respectively. The mean BRS score was 16.6 SD 4.7 and 18.0 SD 1.6 in pregnant and non-pregnant women respectively. The proportion of moderate to severe (stress score 14–40 points) perceived stress was 89% in pregnant women and 53% in non-pregnant women. The proportion of low resilience was 46.7% for pregnant and 21.4% for non-pregnant women. See Table 2.

### Correlation between stress and resilience

Stress and resilience showed a negative correlation to each other (Fig 1). Pregnant women showed higher stress and lower resilience while non-pregnant women contrastingly showed lower stress and higher resilience.

### Comparison of stress and resilience between pregnant and non-pregnant women

**Unadjusted analysis.** Both stress and resilience scores were normally distributed in the sample population. In unadjusted analyses, except PSS item 1 and 6, all PSS item scores were

**Table 1. Background characteristics of study participant, n = 320.**

| Variable | Mean ± SD; n (%) | Pregnant women, n-166 | Non-pregnant women n = 154 | P-value from: t-test, or chi-square |
|---|---|---|---|---|
| **Age** | 27.0±5.7 | 24.7±5.1 | 29.4±5.3 | 0.01[a] |
| **Married** | 295 (92) | 159 (96) | 136 (88) | 0.01[b] |
| **Religion** | | | | |
| Muslim | 195 (61) | 110 (66) | 85 (55) | 0.04[b] |
| Christian | 125 (39) | 56 (34) | 69 (45) | |
| **Average fertility/birth rate** | 2.3±1.2 | 2.0±1.2 | 2.6±1.1 | 0.01[a] |
| **Gestational age, Weeks** | | 22.2 ± 8.1 | - | |
| **Educational status n = 299** | | n = 159 | n = 140 | |
| **Illiterate,** | 25 (8) | 24 (15) | 1 (1) | 0.01[b] |
| Primary (grade 1–8) | 126 (42) | 66 (42) | 93 (66) | |
| Secondary (grade 9–12) | 100 (33) | 44 (28) | 58 (41) | |
| Diploma and above (> grade 12) | 48 (16) | 25 (16) | 23 (16) | |
| **Average family size** | 4.4±1.6 | 3.6±1.4 | 5.2±1.3 | 0.01[a] |
| **Social support, n = 320** | | | | |
| Perceived adequate | 122 (38) | 46 (28) | 76 (49) | 0.01[b] |
| Perceived inadequate | 198 (62) | 120 (72) | 78 (51) | |
| **Current physical activity** | | | | |
| No change | 123 (38) | 48 (29) | 75 (49) | 0.02[b] |
| Increased | 44 (14) | 14 (8) | 30 (20) | |
| Deceased | 153 (48) | 104 (63) | 49 (32) | |

**Note**: a: P-value from t-test; b: p-value form Chi-square test

significantly higher in pregnant compared to non-pregnant women (**Fig 2A**). Numerical values are given in Table A1 in S1 File (student t-test). Linear regression analysis in Table 3 showed that in an unadjusted model pregnancy was associated with higher PSS stress score (β = 4.3; 95% CI: 3.4, 5.3). Similarly, in an unadjusted model except BRS item 2, all BRS item scores were significantly lower in pregnant compared to non-pregnant women (**Fig 2B**). Numerical values are given in Table A2 in S1 File (student t-test). Linear regression analysis in Table 4 showed that in an unadjusted model pregnancy was associated with lower resilience score (β = -3.6; 95% CI: -4.5, -2.6).

**Table 2. Stress and resilience in pregnant and non-pregnant women.**

| Variable | | Mean ± SD; n = 319 | Pregnant women, n-165 | Non-pregnant women n = 154 | P-value (t-test) |
|---|---|---|---|---|---|
| **Stress Z-score** | | -0.0001±1.00 | 0.43±0.89 | -0.43±0.90 | - |
| **Stress raw score (0–40 points)** | | 16.7±4.8 | 18.7±4.3 | 14.4 ±4.3 | 0.01 |
| **Resilience raw score, (6–30 points)** | | 19.4±4.7 | 17.7±3.4 | 21.3±5.2 | 0.01 |
| **Resilience likert scale score** | | 3.2±0.9 | 3.0±0.6 | 3.5 ±0.9 | 0.01 |
| Variables | Category | n (%) | Pregnant women, n-165 | Non-pregnant women n = 154 | P-value (χ2) |
| **Stress level** | Low | 90(28.3 | 18 (11.0) | 72 (46.8) | 0.01 |
| | Moderate | 221(69.5) | 139(84.8) | 82(53.2) | |
| | High | 7(2.2) | 7(4.3) | 0 | |
| | Moderate & high | 228 (71.7) | 146(89.0) | 82(53.2) | |
| **Resilience level** | Low | 110 (34.5) | 77 (46.7) | 33 (21.4) | 0.01 |
| | Normal | 172 (53.9) | 87 (52.7) | 85 (55.2) | |
| | High | 37 (11.6) | 1 (0.01) | 36 (23.4) | |

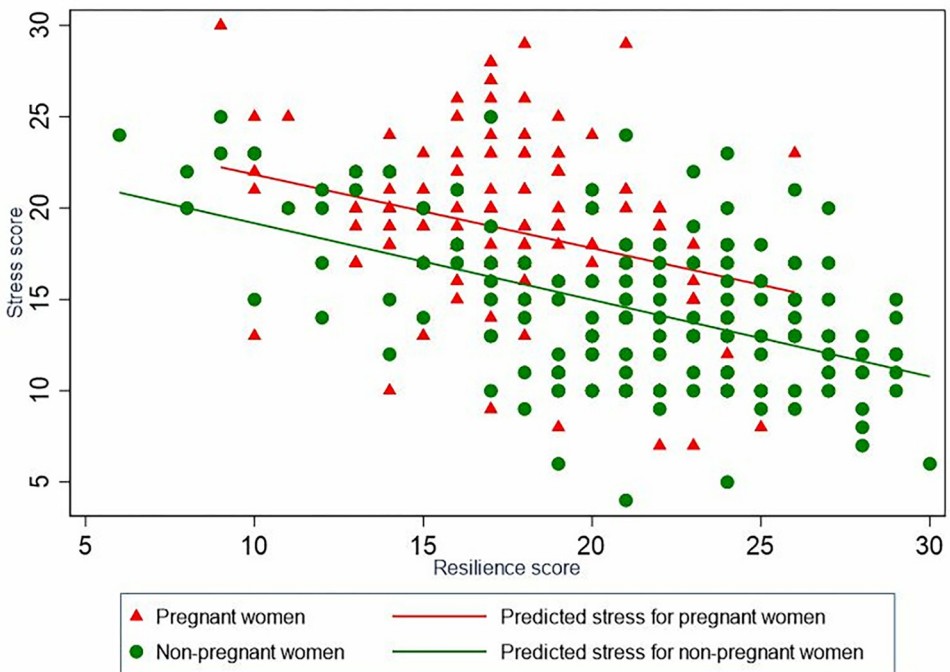

**Fig 1. Line prediction for stress on resilience by pregnancy status.**

**Adjusted analyses.** In a fully adjusted regression model, except PSS items 1, all PSS item scores were significantly greater in pregnant compared to non-pregnant women (**Fig 3A**). Numerical values are given in Table A1 in S1 File (adjusted). For the overall stress score in Table 3, after adjusting for marriage, social support, religion, family size, income, education, occupation, household food insecurity status, distress, and current physical activity level, pregnancy was associated with greater stress ($\beta = 4.1$; 95% CI: 3.0, 5.2). Similarly, except BRS item 2, all BRS item scores were significantly lower in pregnant compared to non-pregnant women (**Fig 3B**). Numerical values are given in Table A2 in S1 File (adjusted). Linear regression

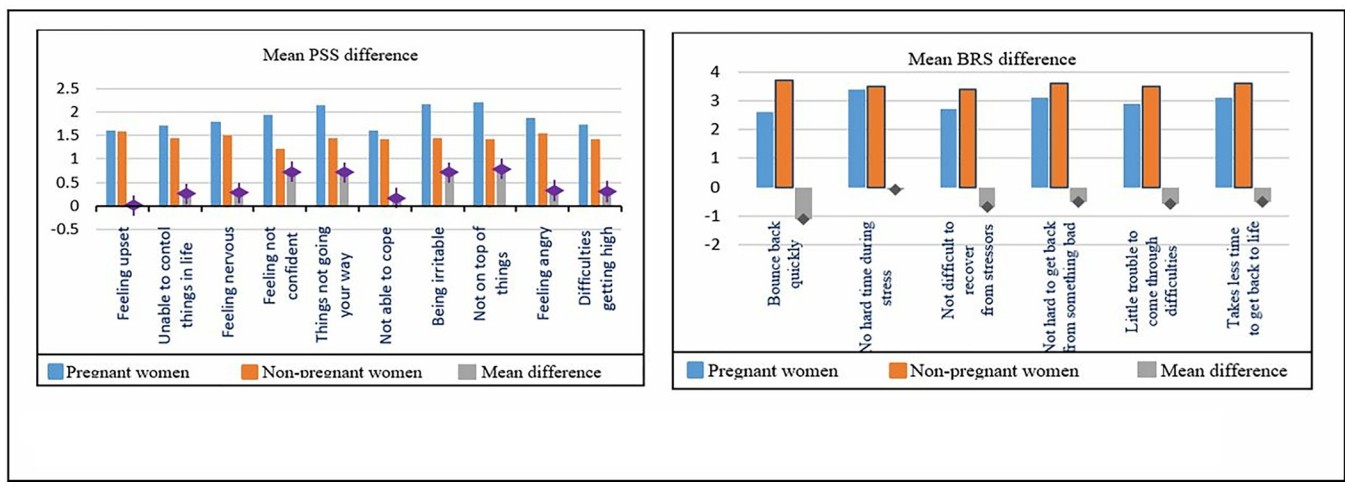

**Fig 2.** Mean difference in (a) Perceived Stress Scale (PSS) items and (b) Brief Resilience Scale (BRS) items between pregnant and non-pregnant women in a simple model.

**Table 3. Association of pregnancy with stress score.**

| Regressed for stress | | Model 1 | | | Model 2 | | | Model 3 | | | Model 4 | | |
|---|---|---|---|---|---|---|---|---|---|---|---|---|---|
| **Exposures** | **Characteristics** | **β** | **[95% CI]** | | **B** | **[95% CI]** | | **β** | **[95% CI]** | | **β** | **[95% CI]** | |
| Pregnancy status | Pregnant | 4.30 | 3.35 | 5.25 | 4.23 | 3.18 | 5.29 | 4.24 | 3.18 | 5.29 | 4.10 | 3.00 | 5.22 |
| **Age** | Age | - | - | - | -0.01 | -0.11 | 0.08 | -0.01 | -0.12 | 0.10 | -0.08 | -0.19 | 0.04 |
| **Parity** | Parity>2 | - | - | - | | | | -.08 | -1.24 | 1.09 | -0.09 | -1.24 | 1.06 |
| **Marital** | Non-married | - | - | - | - | - | - | - | - | - | 0.64 | -1.27 | 2.55 |
| **Social support** | Good support | - | - | - | - | - | - | - | - | - | 0.20 | -0.90 | 1.30 |
| **Religion** | Non-Muslim | - | - | - | - | - | - | - | - | - | 0.23 | -0.77 | 1.24 |
| **Family size** | Family size >5 | - | - | - | - | - | - | - | - | - | 0.11 | -1.12 | 1.34 |
| **Income** | Low income | - | - | - | - | - | - | - | - | - | 0 | | |
| | Medium income | - | - | - | - | - | - | - | - | - | 0.10 | -1.40 | 1.58 |
| | Higher income | - | - | - | - | - | - | - | - | - | 0.32 | -0.93 | 1.57 |
| **Educational status** | No education | - | - | - | - | - | - | - | - | - | 0 | | |
| | Primary | - | - | - | - | - | - | - | - | - | 1.73 | -0.10 | 3.55 |
| | Secondary | - | - | - | - | - | - | - | - | - | 1.25 | -0.69 | 3.19 |
| | College and above | - | - | - | - | - | - | - | - | - | 1.32 | -1.17 | 3.80 |
| **Occupation** | No occupation | - | - | - | - | - | - | - | - | - | 0 | | |
| | Employed | - | - | - | - | - | - | - | - | - | -1.32 | -3.20 | 0.55 |
| | Merchant | - | - | - | - | - | - | - | - | - | -.35 | -1.84 | 1.14 |
| **Household food insecurity** | Household food insecurity | - | - | - | - | - | - | - | - | - | 0.21 | 0.03 | 0.38 |
| **Physical activity** | As usual | - | - | - | - | - | - | - | - | - | 0 | | |
| | Increased | - | - | - | - | - | - | - | - | - | 1.70 | .25 | 3.15 |
| | Decreased | - | - | - | - | - | - | - | - | - | 0.93 | -.13 | 2.00 |
| **Distress score** | Distress score | - | - | - | - | - | - | - | - | - | 0.26 | 0.13 | 0.39 |
| **Substance use** | Ever use | - | - | - | - | - | - | - | - | - | -0.46 | -1.49 | 0.57 |

analysis in Table 4 showed that pregnancy was associated with lower resilience score in the adjusted model (β = -3.3; 95% CI: -4.5, -2.2).

Lastly, in models that adjusted stress for resilience, and vice versa, pregnancy was associated independently with higher stress (β = 2.8; 95% CI: 1.7, 3.9), and with lower resilience (β = -1.6; 95% CI: -2.8, -0.5) (**Fig 4**). Numerical values are given in Tables B1 and B2 in S1 File.

Other factors associated with higher stress in the overall sample include household food insecurity score (β = 0.2; 95% CI: 0.03, 0.4) in model 4, increased level of physical activity (β = 2.2; 95% CI: 0.9, 3.6), decreased level of physical activity (β = 1.3; 95% CI: 0.3, 2.2), distress (β = 0.3; 95% CI: 0.1, 0.4), while resilience was associated with lower stress (β = -0.4; 95% CI: -0.5, -0.3) in model 5. Similarly, household food insecurity (β = -0.2; 95% CI: -0.4, -0.1) and stress were associated with lower resilience (β = -0.4; 95% CI: -0.5, -0.3), while compared to no change in physical activity increased physical activity (β = 2.1; 95% CI: 0.7, 3.5), decreased physical activity (β = 1.2; 95% CI: 0.2, 2.2), and ever use of substance (β = 1.4; 95% CI: 0.4, 2.4) were associated with higher resilience in model 5.

The mutually adjusted model explained 46% of the variance in PSS score and 38% of the variance in the BRS score. The associations between pregnancy with stress and resilience were stable across the different regression models indicating the robustness of the finding.

## Discussion

In this study we compared stress and resilience between pregnant and non-pregnant women. Pregnant women had higher stress and lower resilience than non-pregnant women. The

**Table 4. Association between pregnancy statuses with resilience score.**

| Regressed for resilience | | Model 1 | | | Model 2 | | | Model 3 | | | Model 4 | | |
|---|---|---|---|---|---|---|---|---|---|---|---|---|---|
| Exposures | Characteristics | β | [95% CI] | | B | [95% CI] | | β | [95% CI] | | β | [95% CI] | |
| Pregnancy status | Pregnant | -3.58 | -4.54 | -2.62 | -3.46 | -4.52 | -2.40 | -3.46 | -4.52 | -2.39 | -3.33 | -4.50 | -2.16 |
| Age | Age | - | - | - | -.03 | -0.07 | 0.12 | 0.03 | -0.10 | 0.14 | 0.04 | -0.08 | 0.16 |
| Parity | Parity≥3 | - | - | - | - | - | - | 0.003 | -1.17 | 1.17 | -0.14 | -1.34 | 1.06 |
| Marital | Non-married | - | - | - | - | - | - | - | - | - | -0.08 | -2.07 | 1.91 |
| Social support | Good support | - | - | - | - | - | - | - | - | - | -0.65 | -1.80 | 0.50 |
| Religion | Non-Muslim | - | - | - | - | - | - | - | - | - | -0.66 | -1.71 | 0.39 |
| Family size | Family size >5 | - | - | - | - | - | - | - | - | - | 0.45 | -0.82 | 1.73 |
| Income | Low income | - | - | - | - | - | - | - | - | - | 0 | | |
| | Medium income | - | - | - | - | - | - | - | - | - | -0.74 | -2.30 | 0.81 |
| | Higher income | - | - | - | - | - | - | - | - | - | -0.85 | -2.15 | 0.46 |
| Educational status | No education | - | - | - | - | - | - | - | - | - | 0 | | |
| | Primary | - | - | - | - | - | - | - | - | - | -2.07 | -3.98 | -0.17 |
| | Secondary | - | - | - | - | - | - | - | - | - | -1.76 | -3.78 | 0.27 |
| | College and above | - | - | - | - | - | - | - | - | - | -2.62 | -5.21 | -0.02 |
| Occupation | No occupation | - | - | - | - | - | - | - | - | - | 0 | | |
| | Employed | - | - | - | - | - | - | - | - | - | 1.51 | -0.45 | 3.47 |
| | Merchant | - | - | - | - | - | - | - | - | - | 0.24 | -1.32 | 1.79 |
| Food insecurity | Household food insecurity | - | - | - | - | - | - | - | - | - | -0.32 | -0.50 | -0.13 |
| Distress score | Distress score | - | - | - | - | - | - | - | - | - | -0.02 | -0.16 | 0.12 |
| Physical activity | As usual | - | - | - | - | - | - | - | - | - | 0 | | |
| | Increased | - | - | - | - | - | - | - | - | - | 1.43 | -0.09 | 2.94 |
| | Decreased | - | - | - | - | - | - | - | - | - | 0.83 | -0.28 | 1.94 |
| Substance use | Ever use | - | - | - | - | - | - | - | - | - | 1.58 | 0.50 | 2.66 |

proportion of moderate to severe level of stress was higher in pregnant than non-pregnant women. Similarly, one-third of pregnant women compared to two-thirds of non-pregnant women had normal levels of resilience, suggesting that their coping strategies were hampered during pregnancy. Independent of potential confounders, pregnancy increased stress by 4 points and decreased resilience by 3 points, and the associations with stress and resilience were also largely independent of each other. This finding is robust and associations between pregnancy and stress and resilience are consistent across the different statistical models.

Global data showed that the mean stress score in a sample of 1406 women recruited from different countries was 13.7±6.6 [31] which is lower than our finding in a sample of mixed pregnant and non-pregnant women (16.7±4.8). In the current study, the pregnant women had a much higher mean stress score (18.7±4.3) while the non-pregnant women had only slightly higher mean stress score (14.4 ±4.3) compared to the global data. Similar to our findings, previous studies have documented higher proportions of moderate to severe levels of stress in pregnant compared to non-pregnant women in Thailand [37], Saudi Arabia [9] and Iran [38]. Unlike previous studies, we used a comparative study design to measure association of pregnancy with stress and resilience so that we are able to investigate the independent association of pregnancy with both outcomes. As such, the current study provides more robust findings from a low income setting compared to previous works.

As indicated in previous study [8], various factors could have contributed to the higher burden of stress in this study. Women in general and pregnant women in particular are at a greater disadvantage in LMICs because of a high burden of responsibility with "unpaid care",

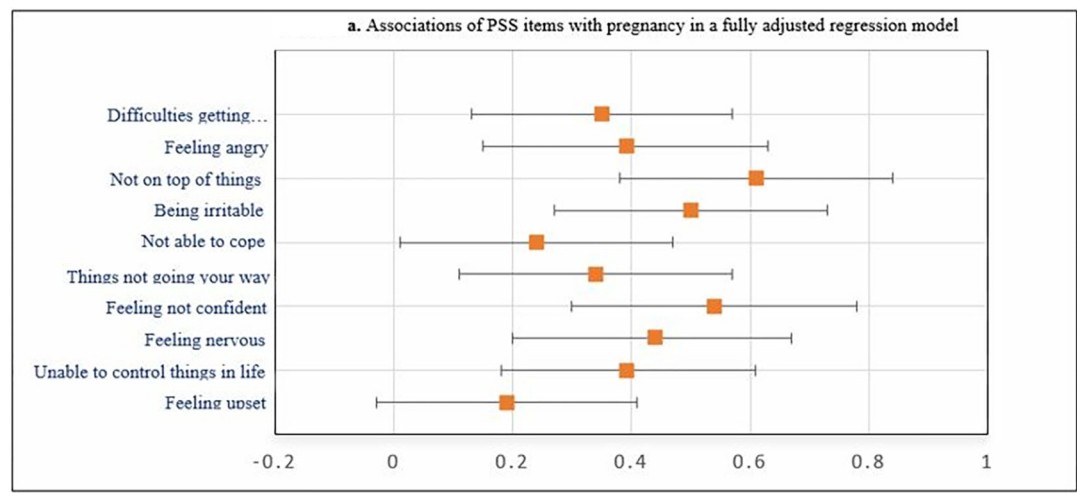

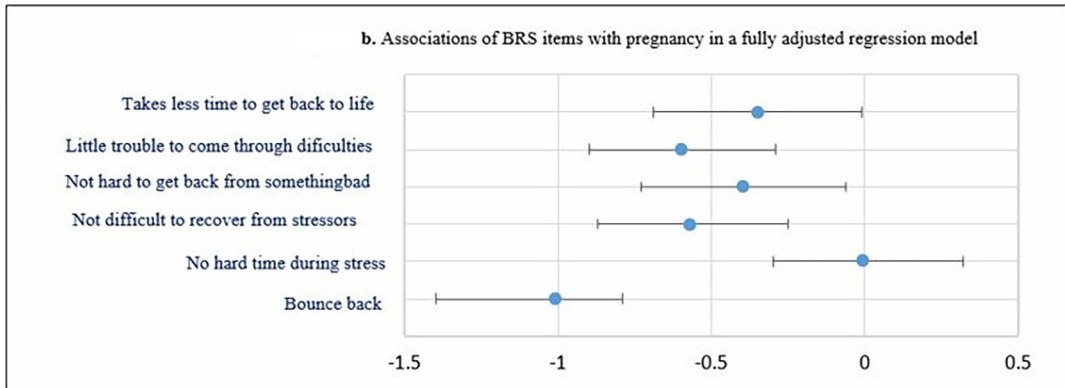

**Fig 3.** Mean difference in (a) PSS-10 item scores and (b) BRS-6 item scores between pregnant and non-pregnant women, in the fully adjusted model.

combined with very low levels of financial and non-financial rewards [10, 39–41]. While life and role transitions to new parenthood and expectations regarding a new baby are sources of personal and family satisfaction, they can also be stressful experiences given uncertainty and concerns over adverse outcomes for the pregnant women and the offspring, which can lead to a range of negative psychological, physical, and mental consequences, all contributing to an increased state of stress [42, 43]. In addition, pregnant women require an increased level of psycho-social support to cope with the pregnancy alongside regular daily responsibilities; but in this study around 72% of pregnant compared to 51% non-pregnant women reported that their social support was inadequate. Moreover in a setting like Ethiopia where maternal mortality is high [44, 45], all pregnancies are considered to be "between life and death" [8] and this situation increases maternal stress significantly during pregnancy. All of these disadvantages that pregnant women encounter on top of their pregnancy are additional life stressors [46–49]. Last, but also important, are the physiological and biological changes taking place during pregnancy such as changes in homeostasis, hormonal levels, body weight, and changes in energy metabolism, all of which can induce stress [50].

Pregnancy is associated with stress and resilience in opposing directions in the current study. While perceived stress score was higher, resilience score was lower in pregnant compared to non-pregnant women. At the first encounter, stressors/adversities affect or challenge an individual's level of resilience or coping strategies and subsequently they progress to induce

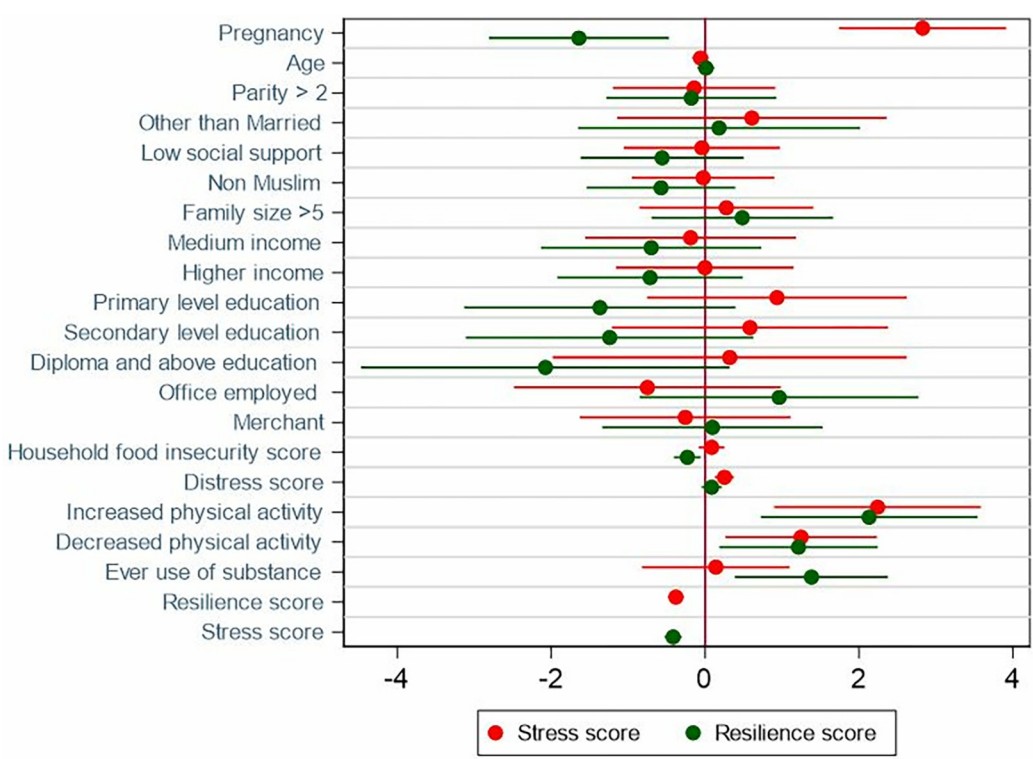

**Fig 4. Association of pregnancy with stress and resilience in the mutually adjusted model.**

stress [4]. While resilience is the ability to absorb shock/stressors, stress is the product of impaired interactions between stressors and coping mechanisms. During immediate exposure to stressors/adversities, resilience helps the body rapidly initiate acute stress responses through activation of the Hypothalamus-Pituitary-Adrenal (HPA) axis, leading to the release of cortisol preparing the body for 'fight or flight' reaction [6]. Our data in both groups (Fig 1) show that there is an inverse association between these outcomes, whereby higher resilience is associated with lower perceived stress. However, each group also showed substantial variability in this association where by pregnant women for the same level of resilience as to non-pregnant women showed higher stress scores (Fig 1).

Normal resilience facilitates a rapid activation of the HPA axis to benefit the body from its activation and release of cortisol followed by a quick culmination of this process [51]. In contrast poor resilience fails to control the continued sustained activation of the HPA axis, leading to a chronic increase in cortisol and resulting in uncontrolled stress [51]. Resilience therefore helps to protect normal homeostasis in the body by enabling it to bounce back from the psychological, physiological and biological effects of stressful situations [3]. Through these mechanisms, normal resilience is mostly inversely associated with pathological stress, anxiety or depression [52]. Thus it is crucial to understand what and what levels of stressors overcome an individual's resilience and what support/intervention types improve resilience so stressors can be overcome. This is especially important for women during pregnancy to benefit both the mother and her offspring, improving the health and productivity of the next generation by breaking the intergenerational transmissions of stress.

The fact that women in general have higher stress score in the current study could indicate the competing responsibility between unpaid household/family responsibility and their aspirations for personal development compounded by an unfavorable educational environment for

females, unrelenting household responsibility and lack of social support making it all difficult to balance their personal development and family life [53, 54] indicating the need for females friendly environment to improve women empowerment in low income settings. Moreover, higher level of household food insecurity, distress and increased physical activity have contributed to the higher stress and lower resilience score in the current study. Household food insecurity is a form of environmental adversities associated with poor mental health outcome [55] and is a common phenomenon in a LMICs. While it is difficult to explain the association between ever use of substance and higher resilience score, this might be attributed to financial access or freedom those women have compared to others. Consistent to previous studies [56, 57] increased level of physical activity is associated with higher resilience but not with lower stress. This could happen because the increase in physically active might be due to the increased household domestic activities making them stressed, or otherwise stressed women might have tried physical activity to reduce their level of stress.

## Strengths and limitations

Our use of a comparative study design, with both groups recruited from a similar setting, enabled us to objectively compare the burden of stress in pregnant and non-pregnant women. In addition, we covered both stress and resilience with additional psychosocial stressors to account for their effect. The limitations in this study include the relatively small sample size, possible selection bias, and selection of unmatched controls. We did not collect and analyze objective stress biomarkers in the current study, hence our results relate only to perceived stress. Moreover, we did not follow the women prospectively to evaluate the longitudinal progression of stress and resilience as well as the impact of the stress on the health and wellbeing of the mother and offspring and on pregnancy and birth outcomes. Lastly seasonal variation might have effect on the finding which is not captured in this study.

## Conclusion and recommendation

Perceived stress is higher and resilience is lower in pregnant women compared to non-pregnant women in Ethiopia. There is a need for more research into the different stress response mechanisms and stress biomarkers during pregnancy. Moreover mechanistic studies and context relevant interventions to improve psychological coping and resilience, and to reduce stress are required so as to improve the health and wellbeing of the mother and her offspring.

## Supporting information

**S1 File.**
(DOCX)

## Acknowledgments

Authors would like to acknowledge study participants.

## Author Contributions

**Conceptualization:** Mubarek Abera, Charlotte Hanlon, Mary Fewtrell, Markos Tesfaye, Jonathan C. K. Wells.

**Data curation:** Mubarek Abera, Hikma Fedlu.

**Formal analysis:** Mubarek Abera, Jonathan C. K. Wells.

**Investigation:** Mubarek Abera.

**Methodology:** Mubarek Abera, Charlotte Hanlon, Markos Tesfaye, Jonathan C. K. Wells.

**Project administration:** Mubarek Abera.

**Resources:** Mubarek Abera.

**Supervision:** Mubarek Abera, Jonathan C. K. Wells.

**Visualization:** Mubarek Abera.

**Writing – original draft:** Mubarek Abera.

**Writing – review & editing:** Mubarek Abera, Charlotte Hanlon, Mary Fewtrell, Markos Tesfaye, Jonathan C. K. Wells.

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
