## [Decision Letter · Decision Letter 0]

31 Jan 2023

PGPH-D-22-01890

Stress and resilience during pregnancy: a comparative study between pregnant and non-pregnant women in Ethiopia

Dear Dr. Abera,

Thank you for submitting your manuscript to PLOS Global Public Health. After careful consideration, we feel that it has merit but does not fully meet PLOS Global Public Health’s publication criteria as it currently stands. Therefore, we invite you to submit a revised version of the manuscript that addresses the points raised during the review process.

We look forward to receiving your revised manuscript.

Kind regards,

Jenil Patel, MBBS, MPH, PhD

Academic Editor

Journal Requirements:

1.Please provide separate figure files in .tif or .eps format.

2. We have noticed that you have uploaded Supporting Information files, but you have not included a list of legends. Please add a full list of legends for your Supporting Information files after the references list. 

3. In the online submission form, you indicated that "Available on request". All PLOS journals now require all data underlying the findings described in their manuscript to be freely available to other researchers, either 1. In a public repository, 2. Within the manuscript itself, or 3. Uploaded as supplementary information.

Additional Editor Comments (if provided):

Reviewers' comments:

Reviewer's Responses to Questions

**Comments to the Author**

1. Does this manuscript meet PLOS Global Public Health’s publication criteria? Is the manuscript technically sound, and do the data support the conclusions? The manuscript must describe methodologically and ethically rigorous research with conclusions that are appropriately drawn based on the data presented.

Reviewer #1: Yes

Reviewer #2: Yes

Reviewer #3: Yes

2. Has the statistical analysis been performed appropriately and rigorously?

Reviewer #1: Yes

Reviewer #2: Yes

Reviewer #3: Yes

3. Have the authors made all data underlying the findings in their manuscript fully available (please refer to the Data Availability Statement at the start of the manuscript PDF file)?

Reviewer #1: Yes

Reviewer #2: No

Reviewer #3: Yes

4. Is the manuscript presented in an intelligible fashion and written in standard English?

Reviewer #1: Yes

Reviewer #2: Yes

Reviewer #3: Yes

5. Review Comments to the Author

Reviewer #1: The paper presents a study of stress and resilience between pregnant and non-pregnant women in Ethiopia. The description of the study rationale, the interpretation of the results and the discussion are balanced. The paper is well-presented with a detailed statistical analysis and may attract wide interest to the readership. I have one minor concern that there is little explanation and discussion about the individual results of the hierarchical logistic analysis, which may decrease the merit of the complicated statistical analysis in this study.

Reviewer #2: This is a well written study to understand if pregnancy is associated with higher stress and lower resilience. It is great to see The associations between pregnancy with stress and resilience were stable across the different regression models indicating the robustness of the finding.

Minor comments:

1) Could higher stress and lower resistance also be associated with many of these families due to poorer education which can possible contribute to unstable households? If this data is available, itd be great if it could be added to the study.

2) Fig 2: labels are overlapping the figure legends and the plots

Reviewer #3: The research work addresses very important aspect. Statistics methods used are proper. The data is collected form a specific area of the country and the data belongs to a limited period, has not spread all round the year. If data of all round the year could be available, then this will make the data more valuable. Month-wise analysis might also be useful.

6. PLOS authors have the option to publish the peer review history of their article (what does this mean?). If published, this will include your full peer review and any attached files.

**Do you want your identity to be public for this peer review?** For information about this choice, including consent withdrawal, please see our Privacy Policy.

Reviewer #1: No

Reviewer #2: No

Reviewer #3: **Yes: **Ramesh Kumar Chandolia

---

## [Editor Report · Decision Letter 1]

22 Feb 2023

PGPH-D-22-01890R1

Stress and resilience during pregnancy: a comparative study between pregnant and non-pregnant women in Ethiopia

Dear Dr. Abera,

Thank you for submitting your manuscript to PLOS Global Public Health. After careful consideration, we feel that it has merit but does not fully meet PLOS Global Public Health’s publication criteria as it currently stands. Therefore, we invite you to submit a revised version of the manuscript that addresses the points raised during the review process.

We look forward to receiving your revised manuscript.

Kind regards,

Jenil Patel, MBBS, MPH, PhD

Academic Editor

Journal Requirements:

2. Please provide separate figure files in .tif or .eps format only and remove any figures embedded in your manuscript file. Please also ensure that all files are under our size limit of 10MB.

Additional Editor Comments (if provided):

The manuscript is scientifically sound and all comments have been adequately addressed. However, there are a lot of grammatical issues in the paper, and the overall write-up needs to be improved. I strongly recommend consulting a scientific writing expert and to proof-read the entire draft to avoid any repititions, word jargon, spelling and grammatical errors.
---

## [Editor Report · Decision Letter 2]

12 Apr 2023

Stress and resilience during pregnancy: a comparative study between pregnant and non-pregnant women in Ethiopia

PGPH-D-22-01890R2

Dear Dr Abera,

We are pleased to inform you that your manuscript 'Stress and resilience during pregnancy: a comparative study between pregnant and non-pregnant women in Ethiopia' has been provisionally accepted for publication in PLOS Global Public Health.

Best regards,

Ahmed Waqas

Academic Editor